# OpenReview forum: "Faithful Simulation of User–Agent–Environment Interactions for Scalable LLM Agent Evaluation"
_ICLR.cc/2026/Conference — Submitted to ICLR 2026_

### Official Review · Reviewer_jYuP · 2025-10-19

**Soundness:** 3
**Presentation:** 4
**Contribution:** 3
**Rating:** 8
**Confidence:** 2

**Summary:**

The paper proposes FUSE, a fully automated framework for simulating User–Agent–Environment interactions to enable scalable and faithful agent evaluation.

The tools are curated from real MCP servers. And the frameworks supports different type of users and environments, which simulates more cases in real-life cases. The task generation pipeline makes the evaluation scalable and is designed to cover diverse tools.

**Strengths:**

- The evaluation considers the different user and environment types.
- The task generation pipeline makes the evaluation scalable and is designed to cover diverse tools.
- The metric is comprehensive and contains procedure, result, and introduce meta-evalutation for the framework itself.
- The systematic experiments uncover critical factors influencing agent performance (environment reliability, user archetypes, trace length), providing concrete best practices for developers (e.g., stress-testing across environments, reporting dual metrics) and guiding the optimization of agentic LLMs.

**Weaknesses:**

- The LLM-as-Judge module may need human verification.
- The Procedure Alignment Score assumes that local semantic similarity implies procedural interchangeability and that severity classifications reflect true risk.

**Questions:**

See the weaknesses.

---

> ### Author Response · Authors · 2025-11-20
>
> We thank Reviewer jYup for their review and for highlighting the systematic experiments (via task generation pipeline), meta-evaluation framework, and the process-alignment metric.
>
> **Q1: Does the LLM-as-a-Judge module need human verification?**
>
> Great question! While FUSE does leverage LLM-as-judge components, it mitigates the high variance of open-ended judging by introducing strong inductive biases, specifically, anchoring the pipeline in real MCP tool definitions and enforcing strict logical dependencies via a deterministic Tool-Relationship Graph.
>
> Nonetheless, we agree with you that relying solely on LLMs is risky. To address this, we are currently finalizing a human alignment study in which (independant) human experts manually label a subset of traces for realism and outcome success. We will calculate the correlation between human labels and our automated metrics and include these statistics in the final version of the paper.
>
> To further investigate the effect of LLM judges, we have extended our evaluation by a Sim-to-Real validation experiment: We used FUSE-generated traces for two domains of the TauBench benchmark [1]  (filtered by our judge and alignment scores), to train specialized Qwen3-4B-Instruct LoRa models. We find that these FUSE-trained models show a performance increase on non-synthetic Tau-Bench tasks (Retail: +2%, Airline: +3% over base), confirming that our evaluation modules are indeed identifying high-quality, valid trajectories that transfer to real-world scenarios, in spite of potential noise introduced by the use of LLM judges.
>
> [1] https://arxiv.org/pdf/2406.12045

---

> ### Author Response · Authors · 2025-11-20
>
> **Q2: Does the Procedure Alignment Score assume that semantic similarity implies local procedural interchangeability? Does severity classification reflect true risk?**
>
> This is an interesting observation regarding the mechanics of our Levenshtein-based metric. We would like to clarify the specific role of these components:
>
> 1. We do not assume that semantically similar tools are functionally interchangeable in the execution sense. Instead, we use semantic similarity as a "soft penalty" mechanism. In traditional Levenshtein distance, a substitution cost is binary (0 or 1). In FUSE, we recognize that if an agent mistakenly calls **filesystem/read_file** instead of **bash/cat** (high semantic similarity), it is a "better" error than calling **database/delete_database** instead of **bash/cat** (low similarity). The metric rewards the agent for identifying the correct intent or functional domain, even if the specific API call was incorrect. This provides a more granular signal than rigid binary matching.
> 2. The insertion cost is weighted by the semantic severity of the side effect. While an LLM-based embedding of tool information cannot perfectly predict "true risk" (which depends on runtime context, e.g., which file is deleted), it functions as a robust proxy for potential risk based on the tool's documentation and definition. This allows us to penalize "reckless" hallucinations (hallucinating a destructive action) much more heavily than "passive" hallucinations (hallucinating a read action).
>
> In summary, the Procedure Alignment Score is not designed to prove functional equivalence, but to measure the divergence of intent between the agent's trajectory and the ground truth graph while respecting potential side-effects.

---

> ### Comment · Area_Chair_4bSv · 2025-11-28
> **Reminder: Engage with Authors During Rebuttal**
>
> Quick reminder: the rebuttal period is still open. Please engage with the authors and update your assessment before the deadline. Thank you for your timely participation.

---

### Official Review · Reviewer_fC5f · 2025-10-24

**Soundness:** 2
**Presentation:** 3
**Contribution:** 3
**Rating:** 4
**Confidence:** 4

**Summary:**

This paper proposes FUSE, a simulation framework for faithful and scalable evaluation of tool-using LLM agents. Its core contribution lies in building a closed-loop user–agent–environment interaction system, addressing the limitations of existing evaluation approaches in terms of cost (manual testing) and authenticity (over-simplified benchmarks).

The main contributions of the paper can be summarized as follows:

1. Task Generation: Leveraging a tool relationship graph, the system automatically generates multi-step task bundles from the MCP server with ground-truth tool sequences.
2. Interaction Simulation: Introduces configurable user prototypes (e.g., planner, information hider) and environment prototypes (e.g., perfect, faulty, adversarial), simulating realistic interaction dynamics.
3. Multi-dimensional Evaluation Metrics: Proposes a process alignment score (a customized edit distance considering tool semantics and risks) and a result success score (end-to-end goal achievement judged by an LLM). These metrics complement each other.
4. Meta-evaluation Framework: Quantifies the trustworthiness of the simulation itself through five faithfulness audit metrics (e.g., solvability, prototype consistency), thereby enhancing the reliability of evaluation results.
5. Extensive Experimental Analysis: Validates the framework with 3,600 runs. Key findings include: environment reliability is the primary determinant of agent performance; user prototypes significantly affect outcomes; there is a moderate correlation between process alignment and result success.

**Strengths:**

The strengths of this paper lie in its high originality, rigorous quality, clear presentation, and notable academic as well as practical value. The core originality of the work is in precisely defining the challenge of evaluating LLM-based agents as a problem of simulation fidelity, and in proposing an integrated solution that creatively combines MCP-protocol-based task generation, configurable user and environment prototypes, and novel process alignment metrics incorporating semantic and risk considerations. Through large-scale experiments (covering 3,600 runs and systematically controlling multiple variables), the authors have achieved a high-quality implementation. Moreover, the paper is clearly structured, logically coherent, and features precise terminology, effectively conveying its technical contributions. Ultimately, the significance of this work lies in providing a principled and scalable benchmark to address a critical bottleneck in the evaluation of LLM agents.

**Weaknesses:**

- As a scalable benchmark, the evaluation covers too few models—only three. It would be more convincing if mainstream large models were all evaluated.
- The task step length does not appear very impressive; existing benchmarks often have longer sequences, which pose a greater challenge to the models.
- Cost concern: Although the paper proposes that the benchmark enables automated evaluation, the number of LLM calls in the experiments seems very large. Moreover, all the models used are top-tier, and the paper does not mention the API costs. Providing cost information and potential, cheaper alternative models would be very helpful for the overall evaluation.
- As a benchmark, the paper does not adequately demonstrate how the benchmark scores correspond to the real-world capabilities of the evaluated models, especially given the heavy use of virtual environments and synthetic data.

**Questions:**

- Can a model be trained? If it is possible to construct the data, then it should also be possible to use the data to train a model and to evaluate it on other benchmarks. If improvements can also be shown there, it would be very convincing.
- Is it possible to conduct a real user study to assess whether the model is producing correct results?

---

> ### Author Response · Authors · 2025-11-20
>
> **Q1: As a scalable benchmark, the evaluation covers too few models—only three. It would be more convincing if mainstream large models were all evaluated.**
>
> We thank the reviewer for this comment and want to first clarify our positioning: we do not intend FUSE to serve as a static benchmark for ranking LLMs, but as a dynamic testing methodology for concrete agent deployments in specific environments. Traditional benchmarks such as BFCL and TauBench rely on fixed toolsets and static task suites, which are ideal for screening general model capabilities. In contrast, FUSE targets system-level behavior in evolving, domain-specific tool ecosystems (e.g., changing GitHub or internal API integrations), where re-building human-curated benchmarks for every new tool or version is prohibitively expensive.
>
> From this perspective, the primary object under test is the AI composite system (agent implementation, tool descriptions, error handling, user persona, etc.), rather than solely the base model. That said, we agree that showing results for more models improves the empirical picture, and we expanded our experiments to include gpt-5.1 and glm-4.6 in the new revision, subject to computational budget constraints and updated the paper to include these new experiments. We will also explicitly position FUSE compared to established benchmarks: use BFCL/TauBench (and similar) to assess general model intelligence, and FUSE to stress-test the deployed agent stack in its actual environment.

---

> ### Author Response · Authors · 2025-11-20
>
> **Q2: The task step length does not appear very impressive; existing benchmarks often have longer sequences, which pose a greater challenge to the models.**
>
> We appreciate this observation. Our design goal with FUSE is not to maximize trajectory length per se, but to faithfully simulate realistic user-agent-environment interactions for a given tool ecosystem, similar in spirit to integration tests in software engineering. Traditional multi-turn benchmarks often emphasize long trajectories as a proxy for difficulty, but they remain tied to fixed toolsets. FUSE instead offers a way to automatically generate synthetic test cases for arbitrary custom agent environments, including their current tools and configurations.
>
> In this sense, FUSE is not competing with long-horizon capability benchmarks; it is providing a system-level testing framework that can be tuned to the deployment context. Longer workflows are straightforward to express in our framework when the target environment naturally induces them (for instance, complex multi-step debugging or multi-API workflows). To address the reviewer’s concern more directly, we will add an ablation study where we extend traces up to 16 function calls for one of the models, demonstrating that (i) the framework scales to longer interaction sequences and (ii) the same methodology applies regardless of horizon length. We will also clarify in the text that FUSE is intended to complement traditional benchmarks: static suites to stress long-term reasoning in a fixed setting, and FUSE to validate the robustness of the actual agent configuration over the lifecycle of system development.

---

> ### Author Response · Authors · 2025-11-20
>
> **Q3: Could you provide cost information and cheaper alternative models?**
>
> The reviewer is correct, API costs were our main limiting factor for not running more experiments and throughout testing in different settings, including different domains, models and task lengths. For example, our latest run with gpt-5.1 for all task lengths specified in the paper with default settings costs around 430 USD: 300 USD being the agent model itself (gpt-5.1), 100 USD being costs for environment&user simulation (gpt-4.1-mini), 30 USD being costs on outcome success. We have added detailed cost breakdown to the manuscript, including distribution for framework components. Overall budget for all experiments from main section are estimated to be around 1300 USD.

---

> ### Author Response · Authors · 2025-11-20
>
> **Q4: As a benchmark, the paper does not adequately demonstrate how the benchmark scores correspond to the real-world capabilities of the evaluated models, especially given the heavy use of virtual environments and synthetic data.**
>
> Again, we frame this work more in the direction of integration testing methodology for AI agents and not fixed benchmark. Nevertheless this potential discrepancy between simulated environments and users and real ones is a big concern. We tried to address it by further experiments with generating data using our benchmark and using it to improve scores of qwen3-4b-instruct over tau-bench using only tool names & descriptions & parameters and not actual implementation. Please find further details below.

---

> ### Author Response · Authors · 2025-11-20
>
> **Q5: Can a model be trained? If it is possible to construct the data, then it should also be possible to use the data to train a model and to evaluate it on other benchmarks. If improvements can also be shown there, it would be very convincing.**
>
> We thank the reviewer for this critical suggestion. We agree that the ultimate test of our data construction framework’s validity is whether the generated data can train a model to perform better on external benchmarks. To address this, we conducted a new set of experiments to validate the transferability of our synthetic data. We used our framework to generate training samples and fine-tuned a model to evaluate on Tau-Bench (Retail and Airline domains).
>
> Using our framework, we generated 600 synthetic traces using Qwen3-4B-Instruct for all components (User, Agent, Environment, Evaluator) in a closed-loop setting for both Retail and Airline domains separately. We applied a strict filter to ensure data quality, selecting samples with high outcome success (>0.8) and high procedural alignment (>0.5). This resulted in a curated dataset of 247 and 300 high-quality samples for Retail and Airline domains respectively. We trained a LoRA adapter on the base Qwen3-4B-Instruct model using this curated synthetic dataset. We evaluated the fine-tuned model Qwen3-4B-Instruct-lora against the base model Qwen3-4B-Instruct on the standard Tau-Bench Retail and Airline tasks, using reflexion user strategy and Qwen3-4B-Instruct as user simulating model. To ensure statistical significance given the variance in agentic benchmarks, we averaged results over 10 runs. The retail domain base model had a Pass^1 score of 0.20±0.02 and the lora-finetuned model had a Pass^1 score of 0.22±0.02. The airline domain base model had Pass^1 score of 0.25±0.06 and lora-finetuned model had Pass^1 score of 0.28±0.03.
>
> We think that these results provide empirical evidence that our “closed-loop” purely synthetic framework generates valid, learnable signals rather than hallucinations or noise, an idea that is consistent with other work in sim2real transition for AI agents [1]. Despite being trained on fully synthetic data generated by the same base model, the LoRA adapter was able to generalize and improve performance on the external Tau-Bench benchmark. We updated the manuscript to include more details about experimenting methodology.
>
> [1] https://arxiv.org/abs/2511.03773

---

> ### Author Response · Authors · 2025-11-20
>
> **Q6: Is it possible to conduct a real user study to assess whether the model is producing correct results?**
>
> We thank the reviewer for this proposal. We’ll ask human labellers to label random traces for realism, including breakdown over user/environment behavior and evaluation model analysis. We will update the manuscript to include that analysis.

---

> ### Comment · Area_Chair_4bSv · 2025-11-28
> **Reminder: Engage with Authors During Rebuttal**
>
> Quick reminder: the rebuttal period is still open. Please engage with the authors and update your assessment before the deadline. Thank you for your timely participation.

---

### Official Review · Reviewer_h2in · 2025-10-30

**Soundness:** 3
**Presentation:** 2
**Contribution:** 2
**Rating:** 4
**Confidence:** 3

**Summary:**

This work presents FUSE, a scalable system for evaluating LLM agents by simulating various interactions between the agent, a user, and an environment.   The method checks how flexible agents are by developing a lot of "archetypes" for both persons, such as "Impatient" and "Planner", and places, such as "Buggy" and "Adversarial".  There are two new ways to measure how well an agent does its job. The first is the procedure alignment score, which checks how well the agent's action steps match up with a ground truth, and the Second is the outcome success score, which checks how close the agent's ultimate output matches with a ground truth.  The framework also incorporates a meta-evaluation phase to check how well it works.  In 3,600 runs of experiments, the most important determinant in an agent's success was the reliability of the environment.

**Strengths:**

The FUSE architecture in this work facilitates controlled, easy-to-extend experimentation through the simulation of adjustable user and environmental archetypes.  The "Phase 4: Meta-Evaluation" is a substantial advancement, since it introduces a series of audits designed to quantitatively assess the simulation's fidelity, thereby addressing a critical deficiency in previous research. The paper introduces a novel "Procedure Alignment Score," a Levenshtein-based metric that evaluates an agent's process against a ground truth, rather than only its outcome. This approach provides useful insights, including one of the important finding that procedural alignment and outcome success are only slightly related.

**Weaknesses:**

1. The adherence scores for "tricky" user archetypes are extremely low. For example, Improviser with 0.31, Information Hider with 0.03. This indicates the simulated user LLM is not faithfully following its instructions, which undermines the validity of the experimental results for those specific archetypes. You may consider adjusting the prompt to overwrite the LLM “helper” default behavior, or try to simulate a larger number of times and collect the one with a threshold of over a certain level.

2. The entire evaluation pipeline is built by LLMs. An LLM generates the Tool-Relationship Graph, an LLM generates the user goal and environment to fit a sampled path, and an LLM judges the final outcome. This "sim-to-sim" evaluation risks a departure from real-world user behavior and task generation, which is the very problem the paper aims to solve. Considering collecting real-world GitHub or Stack Overflow problems, or any problem from the real sources would make the work more reliable.

3. The faithfulness of the user and environmental behaviors is not clear.  Does a 'Buggy' simulated environment really reflect common API failures? It is not validated against real-world data. The authors could analyze public API documentation and error logs to model realistic failure modes. Also, present more scenarios of the experiments, i.e. error analysis.

4. The current domains and short task horizons limit the generalizability of the findings. I suggest adding more domains to improve the work's generalizability. Some good candidates would be e-commerce (managing a shopping cart), calendaring (resolving scheduling conflicts), or cloud infrastructure (provisioning resources), as these involve multi-step, state-dependent interactions.

**Questions:**

See above

---

> ### Author Response · Authors · 2025-11-20
>
> **Q1: Why are the adherence scores for complex archetypes (e.g., "Information Hider”, or “Improviser”) notably lower? Does this suggest that the simulation of users is not faithful for these archetypes?**
>
> We thank the reviewer for pointing out this discrepancy! We investigated the phenomenon and have decided to adjust our method slightly, to better measure true adherence. Previously we calculated message-level archetype adherence scores: i.e. the input to the classifier was a single message from the user or the environment and a corresponding component archetype description. We found this approach to indeed be limited, as complex archetypes are not expected to behave uniformly for all messages in a single conversation. For example information hider is expected to reveal new information as the interaction evolves, which corresponds to a behavioral shift. So instead of message-level classification we moved to conversation-level message aggregation: for user archetype adherence we extract all user messages from the conversation, combine with the user archetype description and classify if the user behaved according to the archetype. We analogously re-evaluate environment archetypes.
>
> After our revision, conversation-level scores are much more aligned.
>
> ### Conversation-level Archetype Adherence Scores by User
> | User Archetype     | Adherence Score (± std) |
> |--------------------|----------------|
> | Planner            | 0.99 ± 0.01    |
> | Improviser         | 0.73 ± 0.34    |
> | Information Hider  | 0.41 ± 0.39    |
> | Russian            | 0.93 ± 0.23    |
> | Goal-shifter       | 0.98 ± 0.09    |
> | Impatient User     | 0.96 ± 0.16    |
>
> ### Conversation-level Archetype Adherence Scores by Environment
> | Environment Archetype | Adherence Score (± std) |
> |------------------------|----------------|
> | Perfect                | 0.96 ± 0.13    |
> | Buggy                  | 0.87 ± 0.24    |
> | Adversarial            | 0.51 ± 0.35    |
>
> We will update the manuscript with the improved methodology based on your contribution, and move our previous message-level results to the appendix. We thank the reviewer for pointing out this inconsistency.

---

> ### Author Response · Authors · 2025-11-20
>
> **Q2: The framework relies entirely on LLMs for all components (graph generation, goal setting, judging), creating a "sim-to-sim" loop. Does this risk diverging from real-world user behavior, and should the pipeline be grounded in real data sources like GitHub or Stack Overflow?**
>
> We fully share the concern regarding overreliance on "LLM-as-a-Judge," a limitation prevalent in earlier works such as ToolEmu [1]. Consequently, FUSE is explicitly designed to depart from purely open-ended LLM generation. Instead, we introduce strong inductive biases that anchor our synthesis in reality and enforce structural validity through three key mechanisms:
> 1. Real-World Anchoring (MCP Ecosystem): Unlike methods that rely on LLM-synthesized (fantasized) toolsets, we seed our generative pipeline with actual tool definitions from the Model Context Protocol (MCP) ecosystem. This defines the action space based on valid software boundaries, significantly reducing "LLM randomness."
> 2. Structured Trajectories (Tool-Relationship Graph): We utilize an explicit Tool-Relationship Graph to enforce realistic procedural dependencies. This ensures our generative pipeline always traverses a valid graph of logical interactions rather than synthesizing arbitrary trajectories.
> 3. Process-Alignment Metrics: We incorporate objective metrics, specifically our Levenshtein-distance approach, to enforce clear structure. This ensures that evaluation is not solely dependent on LLMs acting as final validators.
>
> While these structural constraints minimize hallucination, we acknowledge that any use of LLM-as-a-judge warrants scrutiny. We have therefore initiated a human alignment evaluation to verify that FUSE’s LLM judges align sufficiently with independent human judgment. We will report these results upon completion.
>
> Now, of course, the ultimate test of a synthetic framework’s validity is whether the generated data can train a model to perform better on external, independent benchmarks that use tools implemented in actual code. To address the reviewer's concern about the "departure from real-world user behavior," we conducted a validation experiment transferring our synthetic data to Tau-Bench (Retail and Airline domains). Using our framework, we generated 600 synthetic traces using Qwen3-4B-Instruct for all components in a closed-loop setting. We filtered for high quality (>0.8 outcome success, >0.5 procedural alignment), resulting in 250-300 curated samples per domain. We then trained domain-specific LoRA adapters on the base model using only this synthetic data. We evaluated the models on Tau-Bench (averaged over 10 runs to ensure statistical significance). The results show clear improvement in both:
>
> ### Pass^1 Results
>
> #### Retail
> | Model           | Pass^1 (± std) |
> |-----------------|----------------|
> | Base model      | 0.20 ± 0.02    |
> | LoRA (Ours)     | 0.22 ± 0.02    |
>
> #### Airline
> | Model           | Pass^1 (± std) |
> |-----------------|----------------|
> | Base model      | 0.25 ± 0.06    |
> | LoRA (Ours)     | 0.28 ± 0.03    |
>
>
> These results provide initial empirical evidence that our "closed-loop" synthetic framework generates valid, learnable signals rather than hallucinations or noise. Despite being trained on fully synthetic data generated by a model of the same size, the adapter generalized to improve performance on an external benchmark. This finding is consistent with emerging research on sim-to-real transition for AI agents [2]. We updated the manuscript to include detailed descriptions of those experiments.
>
> [1] https://arxiv.org/abs/2309.15817
>
> [2] https://arxiv.org/abs/2511.03773

---

> ### Author Response · Authors · 2025-11-20
>
> **Q3: Is the "Buggy" environment faithful to real-world API failures? The current simulation lacks validation against real-world data (e.g., public API documentation or error logs). Can the authors provide more qualitative error analysis or scenarios to demonstrate the validity of these simulated failure modes?**
>
> We thank the reviewer for this suggestion. While modeling specific real-world failure modes (e.g., via error logs) is a great suggestion and a valuable future direction, our current "Buggy" environment serves a distinct purpose: evaluating agentic resilience.
>
> The goal is not to simulate real-world behavior perfectly, but to functionally "stress test" the agent. We focus on the category of error (e.g., transient errors vs. parameter errors) to force the agent to employ specific recovery strategies, such as using another tool or argument self-correction. In this context, synthetic errors provide a valid signal for distinguishing robust agents from brittle ones, even if they do not perfectly mirror specific API logs.

---

> ### Author Response · Authors · 2025-11-20
>
> **Q4: Do the current limited domains and short task horizons restrict the generalizability of the findings? The reviewer suggests adding domains with multi-step, state-dependent interactions - such as e-commerce, calendaring, or cloud infrastructure - to demonstrate broader applicability.**
>
> We agree with the reviewer that increasing the diversity of domains and task horizons is essential for demonstrating generalizability. Our primary limitation for the initial submission was the significant computational cost of running extensive multi-turn agentic evaluations. For context, a single full-scale experiment using our pipeline with GPT-4.1 (for all task lengths specified in the paper and with default parameters) costs approximately 330 USD. Given these costs, we initially prioritized depth in limited domains over breadth. We will add this detailed cost breakdown, including distributions for framework components and task lengths, to the manuscript to provide transparency regarding the resource requirements. However, regarding the technical feasibility of the reviewer's suggestion: our framework is designed to be domain-agnostic. Integrating a new domain is highly efficient - often taking only minutes to register the new MCP server definitions and generate the Tool-Relationship Graph. We have further accepted the reviewer's suggestion and are currently running additional experiments on the Apple Ecosystem domain (including calendar, note sharing, maps, contact management). We will also increase the task horizon for one of the models for the current domain (github+filesystem+web search) to address the concern regarding state-dependent interactions. We will include these results in the final version of the paper to confirm the framework's generalizability.

---

> ### Comment · Area_Chair_4bSv · 2025-11-28
> **Reminder: Engage with Authors During Rebuttal**
>
> Quick reminder: the rebuttal period is still open. Please engage with the authors and update your assessment before the deadline. Thank you for your timely participation.

---

### Official Review · Reviewer_zwEH · 2025-11-01

**Soundness:** 2
**Presentation:** 2
**Contribution:** 1
**Rating:** 4
**Confidence:** 3

**Summary:**

The paper proposes FUSE, a framework for evaluating LLM-based tool-use agents in simulated user–agent–environment loops. It automatically generates multi-step tasks from Model Context Protocol tools and assesses agents using two metrics: Procedure Alignment and Outcome Success. Experiments in filesystem, GitHub, and browser domains show that environment reliability and user behavior strongly affect performance, suggesting simulation fidelity is key for reliable agent benchmarking.

**Strengths:**

Overall, the paper presents a well-structured and systematic attempt to evaluate LLM-based tool-use agents under controlled and reproducible simulated settings. While not entirely novel conceptually, it combines multiple evaluation aspects—process alignment, outcome success, and simulation fidelity—into a unified framework that is technically coherent and empirically validated across several domains.

Problem relevance: Addresses an important and timely question—how to rigorously benchmark LLM agents performing complex tool-use tasks.

Framework design: The proposed FUSE system unifies user, agent, and environment simulation in a closed loop, offering configurability and reproducibility.

Dual evaluation metrics: The use of procedure alignment and outcome success captures both process correctness and goal completion, offering complementary insights.

**Weaknesses:**

Missing comparison with existing multi-turn benchmarks: Well-known prior benchmarks such as BFCL, TauBench, and Tau2Bench already target similar multi-turn or tool-use scenarios. The paper does not provide conceptual or empirical comparisons with these, making it unclear what advantages FUSE actually offers.

Unrealistic and oversimplified task generation: The task generation process based on MCP simulations is too simple to reflect real-world user–environment interactions. For example, GitHub and web tasks in FUSE are far less complex than realistic environments like WebArena, where even strong models still perform poorly; thus, the benchmark may overestimate real-world capability.

Limited multi-turn depth: Although positioned as a multi-turn benchmark, FUSE’s longest trajectories contain only eight steps, insufficient to evaluate long-horizon reasoning or recovery behavior seen in realistic multi-round interactions. GPT-4.1 already attains very high scores on FUSE, suggesting the benchmark may not be sufficiently challenging for current top-tier models. This limits its usefulness for tracking future progress in multi-turn agent evaluation.

Overreliance on LLM-as-Judge evaluation: The framework depends entirely on LLM-based judging without any ground-truth final states or verifiable outcomes, which risks bias and makes the reported results less reliable.

**Questions:**

How does FUSE differ from or improve upon existing multi-turn benchmarks such as BFCL, TauBench, or Tau2Bench in terms of design, task diversity, or evaluation methodology?

Can the authors include results for stronger proprietary models (e.g., GPT-5, Claude-4-Sonnet) and major open-source models (e.g., Qwen, GLM, LLaMA) to demonstrate broader benchmarking coverage?

Why was no ground-truth final state used to validate task completion, and how do the authors ensure that the LLM-as-Judge evaluations are unbiased and consistent?

Can the framework support longer multi-turn trajectories (>8 steps), and if not, how might the authors extend it to capture long-horizon reasoning behaviors?

Since FUSE aims to enable large-scale evaluation, what is the computational cost or efficiency compared with existing benchmarks, and can the authors report runtime or resource metrics?

---

> ### Author Response · Authors · 2025-11-20
>
> We thank Reviewer zwEH for their review and are delighted that they recognize the importance of the problem, our framework design and evaluation. Below we answer the reviewers questions.
>
> **Q1: Is FUSE a benchmark? How does it compare to existing benchmarks like BFCL and TauBench?**
>
> Traditional benchmarks use fixed toolsets. However, real-world agents must operate with constantly evolving, domain-specific tools (e.g., GitHub integrations). Constructing human-crafted benchmarks (like TauBench, BFCL) for every new toolset version is prohibitively expensive and does not scale.
>
> FUSE shifts the paradigm from "testing the model" to "testing the system." We recommend to rely on both static benchmarks and FUSE in a complementary fashion: Use BFCL/TauBench to screen general model intelligence. Use FUSE to stress-test the specific agent implementation, including tool descriptions, error handling, and persona resilience. This is not just about inherent model capability, but an end-to-end system test of the specific AI composite system. We have updated the manuscript to make this more apparent.
>
> While synthetic data inherently differs from human-curated data, FUSE offers the first viable pathway to generate test cases for custom agent environments in an automated way. It is not a replacement for general benchmarks, but a necessary framework for the lifecycle of agent development.
>
> We understand that FUSE presents a novel idea in this sense, that insofar has not been represented in traditional benchmarking approaches. We are currently revising our manuscript to reflect this better.

---

> ### Author Response · Authors · 2025-11-20
>
> **Q2: FUSE’s user tasks based on MCP are not complex enough compared to hand-crafted benchmarks like WebArena, where even strong models still perform poorly. The benchmark may overestimate real-world capability.**
>
> Comparing FUSE to WebArena conflates measuring model capability with tool-use validation in a real end-to-end agent system (see Q1). Both approaches have different goals: WebArena is a "field test" for models targeting open-ended web navigation (high environmental noise). FUSE, on the other hand, is an "integration test" generator for specific agents, hence a much more system-specific approach. Note that e.g. GitHub MCP is not a made-up, niche agent system with unrealistic complexity, but rather a widely-used real-world software (25k stars on GitHub). We agree that WebArena is the more challenging task to measure state-of-the-art model capability, however, FUSE’s goal is not to measure model capability, but a specific agent system’s robustness.
>
> Nonetheless, prompted by this review we wanted to investigate the relative complexity of FUSE-like environments further. For this, we conducted an additional Sim-to-Real validation experiment: We hypothesize that the informational value of a FUSE environment can be determined by training a model on the resulting traces, and measuring performance gains on an external, independent benchmark.
>
> Using FUSE, we generated 600 synthetic traces using Qwen3-4B-Instruct as user, environment and evaluator model for the domains of Tau-Bench. After filtering on FUSE quality metrics, we obtained 250-300 curated samples per domain. We then trained LoRA adapters for Qwen3-4B-Instruct using only this synthetic data (for Retail and Airline respectively). We evaluated the resulting models on Tau-Bench:
>
> ### Pass^1 Results
>
> #### Retail
> | Model           | Pass^1 (± std) |
> |-----------------|----------------|
> | Base model      | 0.20 ± 0.02    |
> | LoRA (Ours)     | 0.22 ± 0.02    |
>
> #### Airline
> | Model           | Pass^1 (± std) |
> |-----------------|----------------|
> | Base model      | 0.25 ± 0.06    |
> | LoRA (Ours)     | 0.28 ± 0.03    |
>
>
> These results show that FUSE-synthesized traces are useful in training and hence improve model performance with the real toolsets. This aligns with emerging research on sim-to-real transition [1] and suggests that FUSE indeed captures critical structural complexity, rather than overestimating capabilities.
>
> We updated the manuscript to include detailed descriptions of those experiments.
>
> [1] https://arxiv.org/pdf/2511.03773

---

> ### Author Response · Authors · 2025-11-20
>
> **Q3: Is the framework overly reliant on LLM-as-Judge? How do the authors ensure that the LLM-as-Judge evaluations are unbiased and consistent?**
>
> We share the reviewer's concern regarding the potential risks of unconstrained LLM-as-a-Judge evaluations. While earlier works (e.g., ToolEmu[2]) pioneered LLM-simulated evaluations, they often relied heavily on fully synthesized environments and open-ended judging. FUSE is explicitly designed to mitigate this reliance by introducing strong inductive biases that anchor the evaluation in reality:
> 1. Unlike methods that rely on LLM-generated toolsets, we seed our pipeline with actual tool definitions from the MCP servers. This constrains the action space to valid boundaries, significantly reducing "LLM randomness."
> 2. We use an explicit Tool-Relationship Graph to enforce logical dependencies. This ensures that the "Ground Truth" is not just a final state, but a valid procedural path.
> 3. We complement the LLM judge with a deterministic Procedural Alignment Score based on Levenshtein distance. This ensures that an agent cannot simply "talk" a judge into a positive evaluation; it must adhere to the structural constraints of the ground-truth graph.
>
> Nonetheless, we acknowledge that the use of LLM-as-a-judge mechanics always warrants additional validation. We have thus started a human alignment evaluation, in which we are validating that the LLM judges in FUSE are sufficiently aligned with (independent) human judgement. We will report our results as soon as the experiments have completed.
>
> [2] ​​https://arxiv.org/abs/2309.15817

---

> ### Author Response · Authors · 2025-11-20
>
> **Q4: Can the authors include results for stronger proprietary models and major open-source models  to demonstrate broader benchmarking coverage?**
>
> Yes. We have expanded our experimental suite to include stronger proprietary agents (GPT-5.1) and open-source models (GLM-4.6). These results are included in the updated manuscript to demonstrate the framework's ability to benchmark the current state-of-the-art. Please also note our comments about cost considerations below.

---

> ### Author Response · Authors · 2025-11-20
>
> **Q5: Can the framework support longer multi-turn trajectories (>8 steps), and if not, how might the authors extend it to capture long-horizon reasoning behaviors?**
>
> Trajectory length is one of FUSE’s configurable parameters. Increasing this parameter value is possible and there are no inherent limitations preventing us from doing it (beyond project budget constraints). We agree that increasing the parameter may reveal more model limitations, however of course, increasing the trajectory length only makes sense if the toolset under examination enables meaningful long-horizon workflows. For the toolsets at hand we found the depth chosen in our evaluation to correspond to a realistic workflow complexity.

---

> ### Author Response · Authors · 2025-11-20
>
> **Q6: Since FUSE aims to enable large-scale evaluation, what is the computational cost or efficiency compared with existing benchmarks, and can the authors report runtime or resource metrics?**
>
> Thank you for bringing this up. We agree that cost is a significant factor, and we are currently extending the manuscript by additional details. As an example, we found that running FUSE for ground truth tool sequence lengths of 2, 4, 6, 8 with 900 user tasks for each category, we spent ~$1300 USD in API costs to run the framework across 5 models in the updated revision. We incorporated these additional details in the updated paper.

---

> ### Comment · Area_Chair_4bSv · 2025-11-28
> **Reminder: Engage with Authors During Rebuttal**
>
> Quick reminder: the rebuttal period is still open. Please engage with the authors and update your assessment before the deadline. Thank you for your timely participation.

---

### Author Response · Authors · 2025-11-26

We would like to thank all reviewers for their detailed and constructive feedback. The discussion has helped us substantially improve the clarity, scope, and validation of the work. Below we summarize the key changes incorporated into the revised manuscript (highlighted in blue):

**(1) Clarified the positioning of FUSE.**

Several reviewers noted ambiguity around whether FUSE should be viewed as a benchmark. We have now revised the abstract, introduction, and discussion to make the intent explicit: FUSE is not a static benchmark for ranking models, but a scalable integration-testing methodology for evaluating concrete agent–tool stacks in dynamic tool ecosystems.

**(2) Improved simulation faithfulness and strengthened meta-evaluation.**

Reviewers raised important questions about realism, archetype faithfulness, and LLM-as-judge reliability. We expanded the meta-evaluation suite accordingly:
- Conversation-level archetype adherence (M3, slightly edited): We replaced the earlier message-level classification with conversation-level aggregation. This resolves the underestimation of complex archetypes (e.g., Information Hider) and yields far more stable adherence scores.
- Human-alignment evaluation (M6, new): Independent human raters assessed outcome success and realism for 10 traces. We report strong correlations with our LLM-based judge and overall high realism scores.
- Sim-to-Real validation (M7, new): We added an external-validity study: FUSE-generated traces were used to train LoRA adapters for Qwen3-4B-Instruct, which then showed measurable improvements on the real TauBench tasks (+2-3%). This provides evidence that our synthetic traces contain meaningful procedural signal and do not simply reflect simulator artifacts.

**(3) Expanded model coverage.**

Following reviewer suggestions, we added evaluations of GPT-5.1 and GLM-4.6. These results are now integrated into the main experimental figures.

**(4) Added long-horizon and multi-domain evaluations.**

To address concerns about task horizon length and domain diversity:
we added a long-horizon ablation up to L = 16, showing stable procedural alignment and smooth degradation of outcome success;
we integrated an additional MCP domain (Apple ecosystem: calendar, notes, maps, email), demonstrating that the framework generalizes across heterogeneous tool surfaces.

**(5) Detailed cost and scalability analysis.**

We now include explicit cost breakdowns for each component of the pipeline, clarifying why certain ablations were constrained and demonstrating that the framework remains practical for routine integration-testing workflows.

---

### Meta-Review · Area_Chair_f33C · 2026-01-06

**Summary:**

This paper introduces FUSE, a fully automated framework for simulating user–agent–environment interactions to enable scalable evaluation of LLM-based agents. Rather than proposing a static benchmark, FUSE is positioned as an integration-testing generator for concrete agent deployments operating over evolving tool ecosystems (via MCP servers). The framework combines automated task generation from a Tool-Relationship Graph, configurable user and environment archetypes, dual evaluation metrics (Procedure Alignment and Outcome Success), and a dedicated meta-evaluation layer to assess simulation faithfulness.

Strengths:

Across reviews, there is broad agreement that the paper tackles a timely and important problem: evaluating agentic systems beyond static benchmarks and expensive human-in-the-loop testing. Reviewers highlight several strong aspects:

1. Treating agent evaluation as an integration-testing problem is novel and practically relevant.

2. The closed-loop simulation, configurable archetypes, and multi-dimensional metrics form a coherent and extensible framework.

3. Evaluating trajectories against a ground-truth procedural graph (rather than outcome alone) provides a useful diagnostic signal.

During rebuttal, the authors strengthened the paper by clarifying positioning (not a static benchmark), adding cost analysis, expanding model coverage, extending task horizons, introducing human-alignment checks, and adding sim-to-real validation, showing that FUSE-generated traces can improve performance on TauBench via LoRA fine-tuning.

Weaknesses:

The main concerns raised by reviewers cluster around validation and generality:

1. The framework relies heavily on LLMs (for user, environment, and judging), raising “sim-to-sim” concerns. The added human-alignment and sim-to-real experiments mitigate but do not fully eliminate this concern.

2. Initial experiments were limited in domains and trajectory length; while rebuttal additions (longer horizons, Apple ecosystem domain) help, evidence of broad generalizability remains limited.

3. Some reviewers initially perceived FUSE as a benchmark; while the authors clarified this in revision, the distinction between system-level testing and capability benchmarking should remain clearly emphasized in the revised version.

After reading the paper, reviews and the author responses, my major concern with the work is about the realism of the simulations (user, environment, tasks), i.e., why did the paper select those user/environment archetypes? On what basis? Without justifications or connections with real-life scenarios, those design choices look ad-hoc. How realistic are those simulated tasks? During rebuttal, the paper adds a human alignment experiment, but only very brief discussions are provided and the scale of evaluation is small (see “we labelled 10 held-out conversations with 4 independent raters (goal success, user realism, environment realism; 1–5 scale).” in line 427). I think the realism of the simulated data and their practical usefulness is unclear. The newly added experiment on Sim-to-Real external validity could only slightly address this issue, because that experiment is only done at a small scale with just one model. I would strongly suggest the paper to expand this experiment and comprehensively show the realism of the simulated data and the value of this simulation framework by (1) justifying the designs (such as user types) in the simulation framework, (2) conducting a larger scale evaluation of human alignment and sim-to-real external validity, (3) perhaps also show what weakness of existing agents this simulation framework could expose that other static benchmarks couldn’t.

Given all these comments, I think the paper would benefit from another round of revision and would recommend rejection.

**Reviewer Concerns:**

Please see the comments above.

**Reviewer Scores:**

I think some of the three reviewers who gave 4 initially might have increased their score to 6 if they had been able to engage. But on the other hand, the reviewer who gave 8 initially (with low confidence) wrote a very short review where comments seem a bit superficial.

---

### Decision · Program_Chairs · 2026-01-26

Reject